# Genome-Wide Analysis of the HDAC Gene Family and Its Functional Characterization at Low Temperatures in Tartary Buckwheat (*Fagopyrum tataricum*)

**DOI:** 10.3390/ijms23147622

**Published:** 2022-07-10

**Authors:** Yukang Hou, Qi Lu, Jianxun Su, Xing Jin, Changfu Jia, Lizhe An, Yongke Tian, Yuan Song

**Affiliations:** 1Ministry of Education Key Laboratory of Cell Activities and Stress Adaptations, School of Life Sciences, Lanzhou University, Lanzhou 730030, China; houyk19@lzu.edu.cn (Y.H.); luq2020@lzu.edu.cn (Q.L.); sujx21@lzu.edu.cn (J.S.); jinx19@lzu.edu.cn (X.J.); lizhean@lzu.edu.cn (L.A.); 2Key Laboratory for Bio-Resources and Eco-Environment of Ministry of Education, College of Life Science, Sichuan University, Chengdu 610017, China; g1020160171@gmail.com

**Keywords:** *FtHDACs*, genome-wide, low-temperature responses, Tartary buckwheat

## Abstract

Histone deacetylases (HDACs), widely found in various types of eukaryotic cells, play crucial roles in biological process, including the biotic and abiotic stress responses in plants. However, no research on the HDACs of *Fagopyrum tataricum* has been reported. Here, 14 putative *FtHDAC* genes were identified and annotated in *Fagopyrum tataricum*. Their gene structure, motif composition, *cis*-acting elements, phylogenetic relationships, protein structure, alternative splicing events, subcellular localization and gene expression pattern were investigated. The gene structure showed *FtHDACs* were classified into three subfamilies. The promoter analysis revealed the presence of various *cis*-acting elements responsible for hormone, abiotic stress and developmental regulation for the specific induction of *FtHDACs.* Two duplication events were identified in *FtHDA6-1*, *FtHDA6-2*, and *FtHDA19*. The expression patterns of *FtHDACs* showed their correlation with the flavonoid synthesis pathway genes. In addition, alternative splicing, mRNA enrichment profiles and transgenic analysis showed the potential role of *FtHDACs* in cold responses. Our study characterized *FtHDAC*s, providing a candidate gene family for agricultural breeding and crop improvement.

## 1. Introduction

Tartary buckwheat (*Fagopyrum tataricum*) is a pseudocereal that belongs to the genus Fagopyrum within the Polygonaceae family. Tartary buckwheat is not only an essential medicinal and edible crop, but also adapted to growing in adverse environments, such as harsh climates and nutrient-poor soils [1]. Because of the high content of bioactive flavonoids (rutin, anthocyanins, and quercetin), Tartary buckwheat is the preferred healthy food for the “three-highs population” (high blood sugar, high cholesterol, and high blood pressure) [2]. Additionally, flavonoids such as quercetin were found to fight against COVID-19 [3]. In recent years, the research of Tartary buckwheat has become increasingly popular, and the genome is constantly being sequenced and annotated [4,5,6,7,8,9]. More studies are being conducted on Tartary buckwheat, especially on the synthesis of bioactive flavonoids. The biosynthesis and accumulation of flavonoids are closely related to the living environment in plants. Tartary buckwheat is thought to originate in the mountainous areas of northwest China, and unique phenylalanine pathways have evolved to both respond to and adapt to cold stress [1].

Inducing environmental changes on histone marks at certain loci are important for studying plant stress responses [2]. Moreover, epigenetic editing is a new way of breeding crops [3]. Histone deacetylases (HDACs) are important epigenetic regulators in eukaryotes and are involved in the deacetylation of histone lysine and arginine residues of the H3 and H4 histone. Furthermore, HDACs are highly conserved in many organisms [4,5]. In plants, histone deacetylation is carried out by three HDAC families: RPD3/HDA1, SIR2, and the plant-specific HD2 family [6]. HDACs are associated with transcriptional repression and gene silencing through the deacetylation of lysine residues. The HDACs lack an intrinsic DNA binding domain and are recruited to target genes through their interacting transcription factors and other large multiprotein transcription complexes. The removal of the acetyl groups from histones by HDACs causes tighter chromatin packing, which weakens the combination of the transcription factors and DNA, and is involved in abiotic plant stress responses [7]. The histone deacetylation site, enzyme, and potential function are summarized, and are then associated with transcriptional activation, histone deposition, and DNA repair [8]. HDACs have been characterized in multiple plants, such as in *Marchantia polymorpha* [9], *Arabidopsis* [10], *Zea mays* [11], *Oryza sativa* [12,13], *Camellia sinensis* [14] and *Gossypium* spp. [15]. The *HDAC* gene family is widely involved in plant growth, development, and stress responses. To continue, the function of HDA6 is involved in the morphogenesis of plant roots [16], hypocotyl elongation [17], flowering [18,19], and senescence [20]. However, the basic information and mechanism of *HDAC*s in *Fagopyrum tataricum* remain unclear. Alternative splicing events in *FtHDACs* are also an important aspect of understanding Tartary buckwheat adaptation.

In this study, 14 *HDAC*s were first identified from *Fagopyrum tataricum*, and they were then comprehensively analyzed through the phylogenetic classification, gene structure and chromosomal location, domain organization, *cis*-acting elements, intraspecific collinearity, protein 3D structure, alternative splicing events, and subcellular localization. In addition, the gene expression patterns of *FtHDAC*s were studied during different developmental stages and cold treatments. Here, a fundamental understanding of *FtHDAC*s is provided for Tartary buckwheat growth, development, and cold stress responses, even in the flavonoid synthesis pathway. These results provide information on histone deacetylation in *Fagopyrum tataricum*, while providing an essential candidate gene family for crop improvement.

## 2. Results

### 2.1. Identification and Classification of FtHDACs Genes

In the present study, 14 *FtHDAC* genes were identified in Tartary buckwheat through the BLASTp methods, including nine *RPD3/HDA1s*, two *SIR2s*, and three *HD2s*. The results showed the basic information of the gene family, including renaming, genome serial number, CDS lengths, protein length, amino acid number, and molecular weight, equipotential point, Aliphatic index, and GRAVY (Table 1). In the *FtHDAC* family, the protein length is between 183aa (FtHDA5) and 485aa (FtHDA6-2), and the MW is between 20.68 (FtHDA5) and 56.18 kDa (FtHDA19). The predicted p*I* is between 4.11 (FtHDT2) and 9.39 (FtHDA5). The aliphatic index was 46.23 (FtHDT3) to 119.34 (FtHDA5). The predicted average hydrophilic coefficient (GRAVY) showed that FtHDA5 and FtHDA14 are hydrophobic proteins, and the others are hydrophilic proteins.

Alignments of the full-length *FtHDAC* sequences were used to generate an unrooted phylogenetic tree by MEGA7.0 software. The conserved structure and conserved domain distribution was analyzed by the SMART and GSDS methods. The results showed that the *FtHDAC* family is divided into three subfamilies with a typical subfamily domain. For the largest subfamily, *RPD3\HDA1*, all members have the conserved histone deacetylation functional domain Hist_deacetyl, and each C terminal has the conserved glycine and histidine aspartic residues. The *FtSIR2* subfamily contains two genes, both of which have functional SIR2 domains. The FtHD2 family contains three members, all of which have conserved functional structures in the SCOP domain (Figure 1A).

The secondary structures of FtHDACs are comprised of an α-helix, extended chain, and random coil. The FtHDT1 and FtHDT3 proteins had a large proportion of random-coiled amino acids (>55%), followed by less α-helix (<20%). However, the α-helix of FtHDA5 accounted for 41.53% of the total. Individual proteins showed different secondary structural properties (Appendix A). The predicted 3D structures are shown in (Figure 1B). The structures of FtHDA6-1, FtHDA6-2, and FtHDA19 were similar, suggesting a shared functionality. FtHD1, FtHD2, and FtHD3 were structurally similar, as were FtSRT1 and FtSRT2, illustrating their functional redundancy (Figure 1B).

### 2.2. Conserved Protein Structure and Cis-Acting Element Prediction

The motifs of the three subfamilies of FtHDAC were analyzed by MEME analysis to identify the putative motifs of the HDAC subfamilies in *Fagopyrum tataricum*, and all members contained one distinct motif for each subfamily, verifying that they belonged to the same subfamily, which further provided evidence for the classification of subfamilies (Figure 2A). The result is the same as the phylogenetic tree analysis, most of the closely related members share common motifs.

*Cis*-acting elements analysis showed that FtHDACs are mainly involved in hormone response elements (abscisic acid responsive element, MeJA responsive element, auxin responsive element, and salicylic acid responsive element, gibberellin responsive element), stress response elements (light responsive element, low temperature responsive element, drought responsive element, anoxic specific responsive element, defense and stress responsive element), and functional control elements (flavonoid biosynthetic regulation element, meristem expression regulation element, circadian control element, endosperm development regulation element) (Figure 2B). In addition, the MYB and MYC binding sites were also found, suggesting that the *FtHDAC* family might be regulated by specific transcription factors.

### 2.3. Chromosomal Localization, Phylogenetic Analysis, and Analysis of Gene Duplication Events

The location of the *FtHDACs* gene family in the chromosome and the gene density and gene duplication events among eight chromosomes was explored. Except for the second chromosome, the *FtHDACs* distributed in the other seven chromosomes. Most of the distribution was found on chromosome 6 with four HDAC genes (Figure 3A). The two duplication events were presented in FtHDA6-1, FtHDA6-2, and FtHDA19 to detect the evolutionary relationship between them. The Ka/Ks of FtHDA6-1 was counted by calculating the parameters Ks, Ka, and Ka/Ks ratio in TBtools. The numerical value was 0.11, indicating that the driving force between the evolution of the two genes was mainly purifying selection. To understand the evolutionary relationships of FtHDACs with the HDACs of other plants, a comparative analysis was conducted on the HDAC genes in *Fagopyrum tataricum*, *Arabidopsis*, soybean, tomato, hairy fruit poplar, sweet orange, and rice by MEGA7.0 to construct a phylogenetic tree. The results showed that the HDAC family is highly conserved among the selected species. Most of the genes of the *Fagopyrum tataricum* RPD3\HDA1 subfamily cluster were more closely related with dicots, such as sweet orange and tomato, while the other two subfamilies are basically consistent with monocot dicots, suggesting that *FtHDAC* genes present a different evolutionary history and pathway (Figure 3B).

Furthermore, to explore the underlying evolutionary mechanisms of the *FtHDAC* family, five representative angiosperm species, including some plants from legumes, Chenopodiaceae, cruciferas, Pedaliaceae, and Poaceae, were selected to construct comparative syntenic maps with *Fagopyrum tataricum*. By conducting an interspecific collinearity visualization of major cash crops, such as grasses, maize, and sorghum, the results showed that the *FtHDAC* family exhibited more collinearity with legumes, indicating that *HDACs* are more evolutionarily related between these species. However, it should be noted that they are also closely related to quinoa, roidiaceae, and hemaraceae. The *FtHDAC* family lacked correlation with maize sorghum, and they were not collinear (Figure 4).

### 2.4. Tartary Buckwheat Multiple Tissue Transcriptome Analysis

Using the Tartary Buckwheat Database (TBD), the relative expression of *Fagopyrum tataricum FtHDACs* and part of the flavonoid synthetic genes in roots, stems, leaves, flowers, and three fruit developmental stages was analyzed (Figure 5A, Appendix A). The *FtHDACs* gene family and flavonoid-synthesis-related genes presented different expression patterns. For example, *FtHDT2* showed a high expression level in each tissue but a reduced expression at day 19 of fruit development (Figure 5A). Moreover, *FtHDA19* had a high expression in the roots, stems, and flowers, suggesting some function in these tissues. The flavonoid synthesis-related genes showed different expression patterns during Tartary buckwheat development (Appendix A).

The correlation analysis between the *FtHDAC* family genes and flavonoid synthesis genes was performed (Figure 5B). There are many key enzymes in the flavonoid synthesis pathway, playing with different roles (Appendix A). For instance, the PAL enzyme catalyzes phenylalanine to produce cinnamic acid and coumaric acid plasma, a key enzyme linking phenylpropane compounds and primary metabolism, and it is important roles for regulation of flavonoid compound synthesis. The C4H enzyme is a single oxidative enzyme of the CYP73 series in plant cytochrome P450, as the second enzyme in the plant phenylalanine metabolism pathway, and it catalyzes the synthesis of coumaric acid by the substrate cinnamic acid, with high catalytic vitality. The 4CL enzyme, the last enzyme in phenylalanine metabolism in plants, catalyzes the generation of coumaric acid to COA ester. The CHS enzyme is a member of the polyketide synthase family, which is the first enzyme in the flavonoids synthesis pathway that is closely related to flavonoids and isoflavone synthesis, and an important rate-limiting enzyme in the synthesis pathway. The CHI enzyme is also a key enzyme in the flavonoids synthesis pathway that catalyzes the stereo-isomerization of the chalcone to synthetize the associated (2S) -flavanones. There are 48 key enzymes encoding genes involved in the flavonoid metabolism pathway in buckwheat [21]. The other enzymes mentioned in Figure 5B function in a branch of the flavone synthesis pathway. For example, DRF, ANS, ANR, LAR, F3′H and F3′5′H are key enzymes regulating anthocyanin synthesis. FLS is involved in the synthesis of quercetin and rutin. The results showed the significant negative correlation between *FtHDA2*, *FtHDA8-2*, and four *CHS* (FtPinG0008131000.01, FtPinG0003701300.01, FtPinG0003701500.01, and FtPinG0003710800. 01), suggesting the role of HDA2/8 in the synthesis of chalcone. Furthermore, there was a significant positive correlation between *FtSRT2* and *FLS* (FtPinG0006907000. 01), *ANR* (FtPinG0007896600. 01), and *LAR* (FtPinG0000053800. 01), indicating FtSRT2 is important for flavonol and flavan synthesis. In addition, *FtHDA15* displayed a significant positive correlation with *F3′H* (FtPinG0008925900. 01), *DFR* (FtPinG0002371500. 01), and *CHI* (FtPinG0002790600. 01). From these results, it can be predicted that the *FtHDAC* gene family is important for regulating both the upstream and downstream effects of the flavonoid synthesis pathway in *Fagopyrum tataricum*.

### 2.5. Alternative Splicing of FtHDACs at Low Temperature Treatment

To further analyze FtHDAC transcription at low temperatures, alternative splicing events were analyzed. Previous transcriptomic data were used for analysis here [22]. The experiment included plants subjected to cold memory (4 °C for 6 h, then at room temperature for 18 h, repeated four times, and then placed at 0 °C for 6 h), cold stock (not acclimated, directly exposed to 0 °C for 6 h), and control groups with normal growth condition. Qualitative statistics analysis of alternative splicing events was performed separately for each sample using the AS profile software, and gene model was predicted by Stringtie software (transcript.gtf). The results found that many types of alternative splicing were present in the different cold treatment in Tartary buckwheat. Among these, the largest number of alternative splicing events was observed in both the TSS and the TTS (Appendix A). Further statistics of the alternative splicing of *FtHDACs* family showed different types of alternative splicing (Appendix A), and the results showed *FtHDA8-1, FtHDA8-2*, *FtHDA2*, and *FtHDA9* differ in Control vs. Memory and Control vs. Shock, while *FtSRT1* and *FtHDA14* differ in Control vs. Memory, but not significantly. Moreover, we analyzed whether the low-temperature responses of *FtHDACs* might instead be occurring through the differential splicing of exons. The results showed that the A5SS (alternative 5′ splice site) of *FtHDA8-2* occurred under cold shock treatment (S group) (*p* = 0.019) (Figure 6A), but there was no significant difference (*p* = 0.089) under cold memory (M group) (Figure 6B), suggesting the alternative splicing forms of *FtHDAs* respond to different low-temperature conditions. The alternative splicing was confirmed by PCR analysis with *FtHDA8-2* ORF primers (Appendix A).

### 2.6. Subcellular Localization of the FtHDACs

To determine the location of *FtHDACs* functions, the transient gene expression experiment was performed. We constructed vectors for pCAMBIA1300-FtHDA6-1-eGFP, pCAMBIA1300-FtHDA2-eGFP, and pCAMBIA1300-FtHDT2-eGFP fusion proteins, which were infiltrated into tobacco leaves with the *Agrobacterim tumefaciens* strain GV3101. Three lines of transiently transformed tobacco were obtained, including OE-*FtHDA6-1*, OE-*FtHDA2*, and OE-*FtHDT2*. Through subcellular localization and DAPI staining, the results showed that all five genes were located in the nucleus. Additionally, the light fluorescence signal was found on the cell membrane. (Figure 7).

### 2.7. Low-Temperature Resistance Analysis of Dingku 1

Our previous studies identified Dingku 1 as a freezing-resistant variety [21], and the role of *FtHDAC* in low-temperature resistance will be further explored here. First, the three cold test groups were established: the control group (23 °C), cold memory group (priming: 4 °C for 6 h, followed by 21 °C for 18 h, repeated for 4 days, then 0 °C for 6 h), and cold shock (0 °C for 6 h directly, without priming). The transcriptome and metabolome data of the three cold test groups were extracted and correlation analyses were performed (Appendix A and Appendix A). From the results of the joint analysis, an interesting phenomenon was determined where the different HDACs had completely opposite correlations with the metabolites. For example, FtHDA14 and FtHDA2 exhibited a negative correlation with chloride, vitexin, anthocyanin, apigenin, and kaempferol, which is exactly the opposite of FtSRT1, FtHDA19, FtHDT2, and FtHDA8 (Appendix A). This is also the same in the analysis of the correlation of FtHDAC with transcription factors. For instance, FtHDA14, FtHDA2, FtHDA6-1, and FtHDA9 were exactly the opposite of FtHDA8-1, FtSRT1, FtHDA19, FtHDT2, and FtHDA8-2 for the correlation with the *MYB*, *BHLH*, *NAC*, and *WRKY* family genes (Appendix A). These results illustrated the functional differentiation of *FtHDAC* family genes, and the potential role played in cold responses and tolerance.

To further explore the role of *FtHDACs* in low-temperature responses, 2-week *Fagopyrum tataricum* Dingku 1 seedlings were treated at −4, 0, 4, 8, 12, and 16 °C for 3 h, then recovered at 23 °C for 36 h. Their fresh weight, electrical conductivity, SOD, MDA, anthocyanin, and flavonoids were measured (Appendix A). The results showed that seedlings presented more damage in −10 and −4 °C, which was reflected in the higher ion leakage and MDA, lower SOD, and fresh weight. The contents of anthocyanins and flavonoids of the Dingku 1 seedlings were the highest in the 0 °C treatment, indicating that the suitable low-temperature treatment can stimulate and accumulate flavonoid production to protect the activity of plant cells from the cold. Phenotypic analysis showed a greater effect on the growth of Tartary buckwheat below 4 °C (Figure 8A). Moreover, histone H3 acetylation levels responded to low-temperature stress, and the H3 acetylation levels of Dingku1 were measured at different freezing temperatures by Western blotting. The results showed that H3 acetylation levels increased significantly with the fall in temperature (Figure 8B), and under −4 °C treatment, the total protein was ablated and showed less content.

The expression level of *FtHDACs* was tested by RT-qPCR; cold treatment conditions were the same as described above (Figure 8C). The results showed that the expression levels of *FtSRT1*, *FtSRT2*, *FtHDA5*, *FTHDA6-1*, and *FtHDA6-2* increased significantly as the temperature decreased, indicating that these five genes are positive regulators in Dingku 1 cold responses. However, the expression levels of *FtHDA8-2* and *FtHDT1* decreased significantly as the temperature decreased, indicating that these two genes play a negative regulatory role in Dingku 1 when facing low-temperature stress.

To further investigate the function of *FtHDACs*, we constructed the overexpression lines OE-*AtHDA6* and OE-*FtHDA6-1,* and they were identified (Figure 9B,C). Subsequently, the seedlings of four genotypes were frozen at −10 °C for 2 h then recovered in room temperature for 2 days. The seedlings of *axe1-5* showed the worst tolerance in phenotypes and ion leakage (Figure 9A,D); axe1-5 (also called hda6-6) is a hda6 mutant carrying a point mutation on an HDA6 splicing site, and it is a mutant line commonly used to study the function of HDA6 [23]. The levels of mRNA in *AtDREB1A, AtDREB1B,* and *AtDREB1C* in the different transgenic lines were assessed. CBF (C-repeat binding transcription factor/dehydrate responsive element binding factor, DREB) is the hub of the plant CBF cold resistance pathway, which mainly regulates the expression of a large number of downstream cold resistance genes, which is extremely important for enhancing plant cold resistance ability. Its expression is also induced by other abiotic stresses, such as, drought, salinity, mechanical injury, and osmotic pressure. The results showed the expression of *CBF* increased dramatically in the cold treatment, especially in OE-*AtHDA6* and OE-*FtHDA6-1* lines. Interestingly, the expression of *AtDREB1* reduced in *axe1-5* when compared to the other lines (Figure 9E,F), and *AtDREB1C* gave the most significant performance (Figure 9G).

## 3. Discussion

Epigenetic studies include DNA methylation, histone acetylation, ubiquitylation, phosphorylation, and intracellular non-coding RNA regulation. These changes in chromatin structure determine the gene expression by activating or silencing, thus adapting to the external growth environment [24]. Histone acetylation modification is jointly regulated by histone acetyltransferase (HAT) and histone deacetylase (HDAC) [25]. Histone lysine acetylation is an important chromatin modification for the epigenetic regulation of gene expression in response to environmental stress [26]. Histone deacetylation affects many growth and developmental events in plants, such as the flowering stage, embryogenesis, root hair development, abscisic acid, and salt reaction [27,28]. All histone modifications are reversible, which may provide a flexible pathway to for regulating gene expression during plant development and in response to environmental stimuli.

HDAC has been isolated from plants including *Arabidopsis*, rice, maize, soybean, cotton and potato. *Arabidopsis* contains 18 HDACs, which can be divided into three families: RPD3/HDA1, SIR2, and plant-specific HD2 [29]. Among them, the RPD Type 3 HDAC serves to maintain chromatin states and regulate housekeeping gene activity in yeast, *Drosophila*, *elegans*, and metazoans [30]. Members of the RPD3/HDA1 family can be further divided into three categories [31]: Class I, including HAD6, HAD7, HAD9, and HAD19; Class II Group HAD2; Class III, which contains HAD5, HAD15, and HAD18. The others were HAD8, HAD14, HAD10, and HAD17. The HD2 subfamily can be divided into HD2A, HD2B, HD2C, and HD2D [32]. The SIR2 family HDAC is a nicotinamide adenine dinucleotide (NAD)-dependent HDAC, and has two members of SIR2-like HDACs, SIR1, and SIR2 [33]. HDACs are found to localize to membranes, nuclei, or nucleoplasmic shuttling, with different functions depending on their localization. RPD3/HDA1 is the largest subfamily in the HDACs family, and the family depends on Zn^2+^. Members of this family all contain a typical histone deacetylase domain [34]. Its structural analysis found that the HDAC structure of the family members is highly conserved, while the other parts are poorly conserved. Therefore, it may be the main reason for the functional differences between different members of the same family, and the protein activity of this family member can be inhibited by triostatin (TSA).

Here, a total of 14 *FtHDAC* family genes were retrieved from the Tartary buckwheat genome through HDACs conserved domains in *Arabidopsis* (Table 1). It is also composed of three subfamilies, including nine RPD3/HDA1 subfamily genes, two SIR2 subfamily genes, and three HD2 subfamily genes (Figure 1A). The characterization of the MEME motif shows that it conforms to the basic structural compositions of the *HDAC* family genes in plants. The prediction of *cis*-acting elements showed that the *FtHDAC* gene promoters contained hormone responsive elements such as ABA and methyl Jasmonate, light responsive elements, drought responsive elements, and low-temperature responsive elements, suggesting that *FtHDAC* plays an important role in the growth and development of Tartary buckwheat and in coping with various environmental changes (Figure 2). This conclusion is also supported by previous reports in which *HDACs* play a critical role in regulating abiotic stress responses. For example, histone acetylation changes in plant responses to drought stress [35]. *Arabidopsis HDA6* is required for freezing resistance [36] and salt stress [28,37].

The construction of interspecific collinearity can help researchers to better understand the evolutionary relationship between FtHDACs and the HDACs of other plant species (Figure 3 and Figure 4). The secondary structure of FtHDACs is helpful for understanding the mechanism of its actions (Figure 1B). A tissue expression analysis showed that the expression level of the three FtHDT2 subfamily members is relatively higher than other family members during normal development (Figure 5A), especially in the flowers, roots, and stems. It was inferred that these genes in the same clade with similar expression patterns might play similar roles in physiological processes. *AhHDA19* was specifically expressed in the root and stem, and *FtHD19* in Tartary buckwheat is greatly expressed in the root and stem, showing a similar expression pattern, implying they might execute functions dominantly in the root and stem. Furthermore, the *FtHDA2, FtHDA9, FTHDA6-1, FTHDA8-1, FtHDA5,* and *FtRPD3/HDA1* subfamily members were highly expressed in the roots, suggesting that these genes play an important role in root morphogenesis. Correlation analysis revealed the potential relationship between *FtHDACs* and flavonoids synthesis genes (Figure 5B). Seven types of alternative splicing have been observed in both humans and rice (Oryza sativa), including intron retention, exon skipping, mutually exclusive exon, alternative 5′ splicing, alternative 3′ splicing, alternative first exons and alternative last exons [38]. In this study, the type of HDA8-2 alternative splicing is A5SS (alternative 5’ splice site) (Figure 6). Alternative splicing is a possible “molecular temperature” that allows plants to quickly adjust the abundance of functional transcripts to adapt to environmental perturbations [39,40]. A preliminary analysis of subcellular localization was performed, and the results revealed the nucleus localization of FtHDACs (Figure 7). The subcellular localization of target genes occurs through transient transformation. Transient gene expression is an effective experimental tool for research on plant gene function [41]. Figure 7 shows the fluorescent signal that appeared on the membrane, but the function is unknown. *Arabidopsis* HDA6 does have the characteristics of cytoplasmic localization; for example, HDA6 interacts with FLD at the nuclear periphery [42]. BIN2 interacts with HDA6 in the cytoplasm and nucleus [43]. The following low-temperature responses of *FtHDACs* have been discussed. Firstly, phenotype and histone acetylation of the Dingku1 treated at different low temperatures were determined, and the expression of *FtHDACs* was also analyzed in at different low temperatures. Interestingly, the result showed that 16 °C was the optimal growth temperature for Tartary buckwheat. Moreover, the low temperature caused significant changes in the expression of *FtHDACs* when compared to room temperature (Figure 8). A stable transgenic *Arabidopsis* line of *FtHDA6* was constructed to further explore the freezing function of FtHDAC. The results revealed that FtHDA6 showed a trend to improve freezing resistance (Figure 9). Previous studies have conducted a systematic analysis of the cooling tolerance mechanism in Tartary buckwheat [1], and our study fills the gap in the mechanism of epigenetic regulation.

Histone acetylation plays a key role in plant development and the response to various environmental stimuli by regulating gene transcription. It was revealed that Tartary buckwheat HDACs could be classified into three major subgroups: RPD3/HDA1, HD2-like, and SRT, which is similar to *Arabidopsis*. Moreover, FtHDACs also carried the functional catalytic domains and other conserved domains, as well as motifs similar to their counterparts in *Arabidopsis*. The function of FtHDA6 in low-temperature responses and flavonoid synthesis pathways was predicted. In brief, the present study highlights the implication of Tartary buckwheat HDACs in various developmental processes and low temperature stress adaptation. In addition, this study also highlights the potential role of Tartary buckwheat HDACs in flavonoid synthesis pathways. This study provides motivation for the investigation of the biological and cellular functions of histone acetylation, which will eventually lead to the long-term improvement of agronomic characteristics and abiotic stress tolerance in *Fagopyrum tataricum*.

## 4. Materials and Methods

### 4.1. Plant Growth, Cold Treatments, and Tissue Collection

Tartary buckwheat seeds (Dingku 1) were provided from the Qinghai Academy of Animal Science and Veterinary Medicine of Qinghai University (Qinghai, Xining, China). Tartary buckwheat and tobacco seedlings were cultured in greenhouses according to [21]. After 4–8 h of soaking the seeds in ddH_2_O, the seeds were disinfected using a 15% NaClO solution and then placed in a culture dish with two layers of gauze. The culture dish was moved to a greenhouse and cultured until germination. For the cold stress experiment, 2-week-old seedlings were treated in −4, 0, 4, 8, 12, and 16 °C for 3 h, and were all recovered in 23 °C at 36 h. The cold treatment conditions of the transcriptome was mentioned according to [21]. The seedling of the cold memory group (memory) was kept at 4 °C for 6 h, then at room temperature for 18 h, repeated four times, and then placed at 0 °C for 6 h. The cold stock (not acclimated) was directly exposed to 0 °C for 6 h, and control groups experienced normal growth conditions. The leaves and roots of the samples were collected, immediately frozen in liquid nitrogen, and stored at −80 °C.

### 4.2. Genome-Wide Identification of Fagopyrum Tataricum HDACs Genes

The genome sequence of the Tartary buckwheat genome was downloaded from the Tartary Buckwheat Genome Project (http://mbkbase.org/Pinku1/) (accessed on 1 March 2020) [44]. Amino acid sequences of the *Arabidopsis HDAC* family genes were downloaded from the TAIR website (https://www.arabidopsis.org/) (accessed on 1 March 2020), then used as queries in local BLASTP against the Tartary buckwheat genome (e-value = 1 × 10^−10^). Furthermore, SMART (http://smart.embl-heidelberg.de/) (accessed on 1 March 2020) and HMMER (https://www.ebi.ac.uk/Tools/hmmer/search/phmmer) (accessed on 1 March 2020) were used to confirm the presence of the HDAC domain. The physicochemical properties of the *FtHDACs* genes were predicted by the ExPASy website (https://web.expasy.org/compute_pi/) (accessed on 1 March 2020), including the protein size, molecular weight (MW), isoelectric point (p*I*), and aliphatic and GRAVY index. In addition, the gene structure was visualized by GSDS2. 0 (https://gsds.gao-lab.org/) (accessed on 1 March 2020). The genome sequences and amino acid sequence of the *FtHDACs* are presented in Appendix A.

### 4.3. Conserved Protein Structure, Cis-Acting Element Prediction, and Protein 3D Structure Analysis

The conserved motifs of the FtHDACs protein sequences were analyzed via the online Multiple Expectation Maximization for Motif Elicitation (MEME) version 4.11.1 (http://meme-suite.org/tools/meme) (accessed on 1 March 2020) [45], and the maximum number of motifs was set to 18. The 1200 bp upstream genomic DNA sequences were analyzed in the PlantCARE (http://bioinformatics.psb.ugent.be/webtools/plantcare/html/) (accessed on 1 March 2020) database for *cis*-acting element prediction. The data was visualized by TBtools [46]. The secondary protein structure was performed by PRABI (http://www.prabi.fr/) (accessed on 1 March 2020). An automated protein structure building was conducted by the Robetta (https://robetta.bakerlab.org/) (accessed on 1 March 2020) program [47]; the data are showed in Appendix A.

### 4.4. Phylogenetic Analysis, Genome Distribution, and Gene Duplication

The phylogenetic tree of the FtHDAC protein family was constructed in the Neighbor-Joining method via the MEGA7.0 software with 1000 replicated bootstrap values [48,49] and the p-distance and pairwise gap deletion parameters engaged. The chromosomal distribution of *FtHDACs* was built by the TBtools and Itol (https://itol.embl.de/) (accessed on 1 March 2020) [46]. The parameters (Ks-synonymous substitution rate and Ka-nonsynonymous substitution rate) of the duplication events were computed by TBtools. Amino acid sequences used for phylogenetic analysis are listed in Appendix A.

### 4.5. Alternative Splicing Analysis

ASprofile software was used to perform qualitative analysis statistics of alternative splicing events for each sample on the gene model (transcript.gtf), predicted by Stringtie. We performed the alternative splicing events analysis based on the gene structure annotation information of Tartary buckwheat [22]. The rMATS is a computational tool for detecting differential alternative splicing events from RNA-Seq data. Based on RNA-Seq data, rMATS can automatically detect and analyze alternative splicing events corresponding to all major types of alternative splicing patterns [50,51].

### 4.6. Transcriptome and RT-qPCR Analysis

The raw data RNA-seq of *FtHDACs* in different tissues (root, stem, leaf, flower, and three-stage fruit) were retrieved from the Tartary Buckwheat Database (TBD) (http://shujuku.zuotukeji.net/) (accessed on 1 March 2020) (accessed on 1 March 2020) [52,53]. The correlation analysis was visualized via OmicStudio software and Tbtools; the data are shown in Appendix A. The heat map was generated using OmicStudio software (www.omicstudio.cn) (accessed on 1 March 2020). For quantificational real-time PCR, the total RNA was extracted using the RNA prep Pure Plant Plus Kit (Tiangen Biotech, Beijing, China). Then, 2 μg RNA was used for the first strand of cDNA synthesis using reverse transcriptase (Vazyme, R211-02, Nanjing, China). Real-time PCR amplification was carried out with the Bio-Rad CFX96 system using SYBR Green I (Vazyme, Q711-02, Nanjing, China). The reaction system contained 10 μL SYBR Master Mix buffer, 0.4 μL each of the primers (10 μM), 1 μL of template, and 8 μL ddH_2_O. The thermal profile for qRT-PCR was as follows: pre-denaturation at 95 °C for 5 min; cycling stage at 95 °C for 10 s, 60 °C for 30 s, 72 °C for 15 s, 40 cycles; melting stage at 95 °C for 15 s, 60 °C for 1 min, 95 °C for 15 s. Three independent biological replicates were used in the analysis and the 2^−(ΔΔCt)^ method was applied for the analysis gene expression. Here, *FtH3* was used as a reference gene to normalize the expression level. The RT-qPCR primer sequences used in this paper are all listed in Appendix A.

### 4.7. Western Blot Assays

The tested seedling samples were ground, the total proteins were extracted, and specific proteins were detected as described [54]. Histone H3 was used as an equal loading control. The antibody in this study used in Western blotting was anti-H3K9K14K18K23K27ac (ab47915, Abcam, Lot: GR137984-20, Cambridge, UK).

### 4.8. Construction of the Arabidopsis Transgenic Plants and Low Temperature Treatments

The sequence of HDA6-1 CDs was amplified and inserted into the pCAMBIA1300-eGFP vector using a ClonExpress II One Step Cloning kit (Vazyme, C112-02) to generate pCAMBIA1300-FtHDA6-1-eGFP constructs. The true recombinant plasmid was transformed into *Agrobacterium tumefaciens* GV3101, then it was transferred into wild-type *Arabidopsis* by the floral-dip method. The positive transgenic plants were obtained by resistance screening (the selectable marker gene is hygromycin) and quantificational real-time PCR identification. The target protein has been proven in transgenic lines by confocal fluorescence microscopy (ZEISS/LSM880) (Appendix A) [55,56]. The genomic DNA of *Arabidopsis* was isolated from leaves using the Plant Genomic DNA Extraction Kit (TiangenP5008, China). For phenotypes and ion leakage analysis of low-temperature stress experiments in transgenic plants, 2-week-old transgenic seedlings were treated in −10 °C for 2 h, and were recovered in 23 °C at 3 days. For *CBF* gene expressional analysis of low-temperature stress experiments in transgenic plants, 2-week-old transgenic seedlings were treated in 0 °C for 3 days, and total RNA was extracted, cDNA was reverse-transcribed, and subjected to the qPCR assay.

## 5. Conclusions

We performed a comprehensive analysis of the HDAC gene family in Tartary buckwheat. A total of 14 *FtHDAC* genes, including nine *FtRPD3/HDA1*, three *FtHD2s*, and two *FtSIR2s*, were genome-wide identified. The gene structure, chromosomal distribution, motif prediction, *cis*-acting element prediction, phylogenetic relationship, 3D structure, expression patterns, alternative splicing, subcellular localization, and heterologous expression were conducted and analyzed. Moreover, the expression pattern of *FtHDACs* was shown in various tissues/organs to be involved in the developmental process. The *cis*-acting element prediction and RNA-seq data indicated that the *FtHDAC* is involved in low-temperature stress responses and flavonoids’ synthesis. This conclusion was also validated in FtHDA6 transgenic *Arabidopsis thaliana*, which indeed affects plant freezing tolerance. According to the above results, the potential mechanism for the roles of *FtHDACs* in the regulation of flavonoids at low temperatures in Tartary buckwheat was proposed (Figure 10). We first collated and analyzed the *HDACs* in Tartary buckwheat and attempts to discover the biological function of *FtHDACs*. This result implies that chromatin regulations are important for low-temperature tolerance and the flavonoid synthesis of Tartary buckwheat.

## Figures and Tables

**Figure 1 ijms-23-07622-f001:**
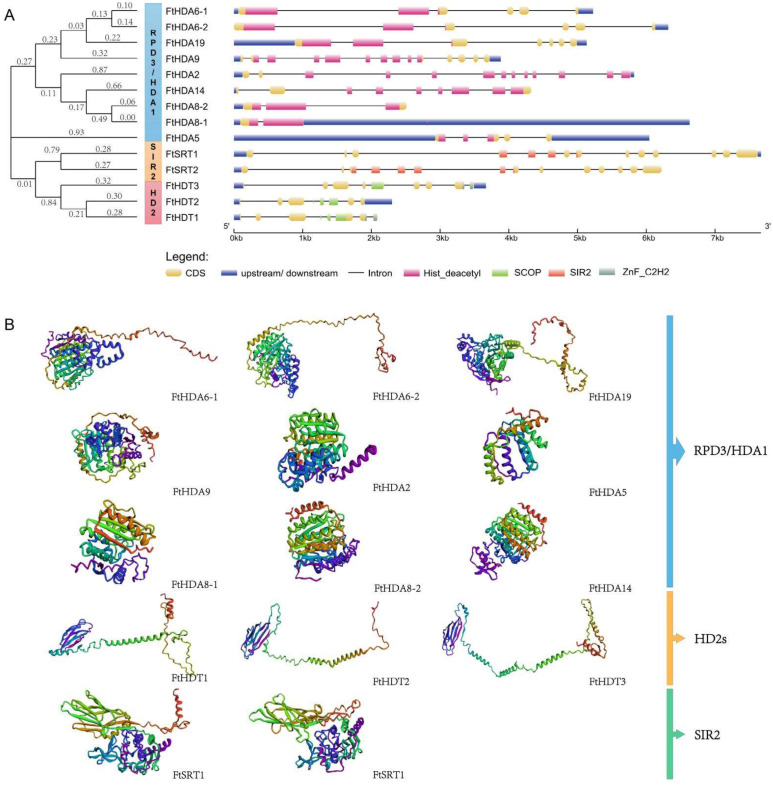
Phylogenetic relationships, motif structure and three-dimensional protein structure of FtHDACs. (**A**) Phylogenetic relationships and exon/intron structures of *FtHDAC* genes. The different structural units are represented by different colors, respectively. (**B**) Models and ribbon diagrams show the 3D domain structure of 14 FtHDACs proteins. Blue to red, N- to C-terminus.

**Figure 2 ijms-23-07622-f002:**
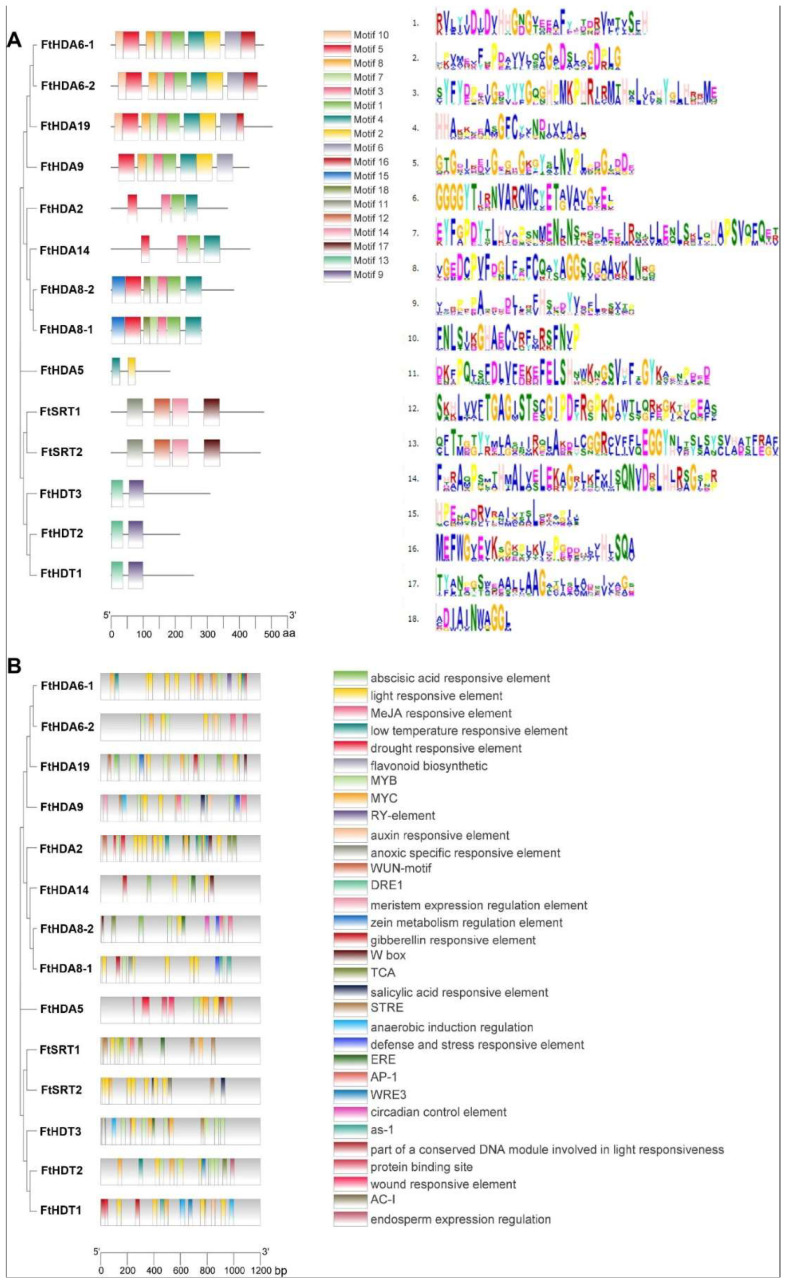
Conserved protein structure and *cis*-acting element prediction of FtHDACs. (**A**) Conserved motifs of FtHDACs proteins using MEME, and details of the 18 conserved motifs shared among the FtHDACs proteins. Each motif is indicated by a colored box numbered on the right. The length of the motifs in each protein is shown as a proportion. Motif symbol and motif consensus also are shown. (**B**) *Cis*-acting element prediction via PLANT CARE. There are 32 types of cis-elements, marked with rectangles of different colors.

**Figure 3 ijms-23-07622-f003:**
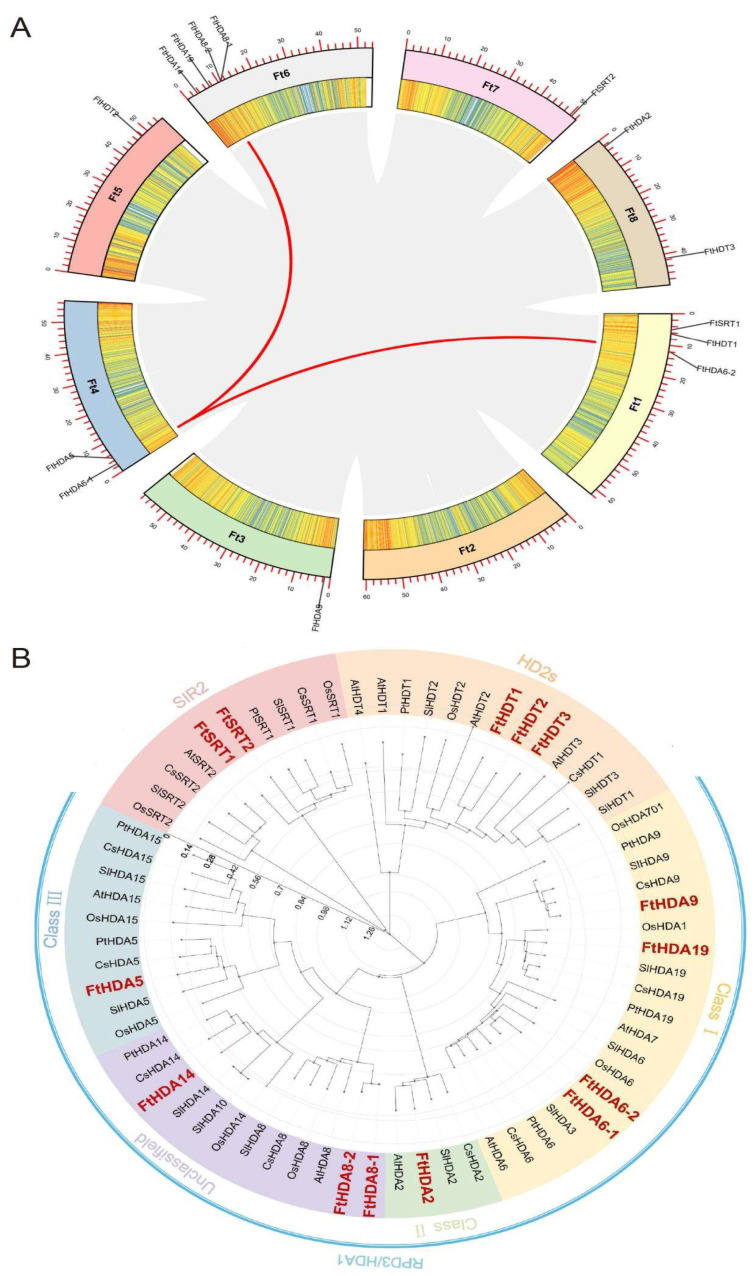
Chromosomal localization and Phylogenetic analysis of *FtHDA6*. (**A**) Distribution of *FtHDACs* gene on *Fagopyrum tataricum* chromosomes, intraspecies collinearity and gene density. (**B**) Phylogenetic tree construction of *HDACs* gene family from *Fagopyrum tataricum* (Ft)*, Arabidopsis* (At)*,* Solanum lycopersicum (SI), Oryza sativa (Os), Populus trichocarpa (Pt), Citrus sinensis (Cs). Each subgroup ID number is in the outer circle of the phylogenetic tree and branches with less than 70% bootstrap support are collapsed (replicated 1000 times). The different filling colors indicate different gene subfamilies, and the length of the clade indicates the evolutionary distance.

**Figure 4 ijms-23-07622-f004:**
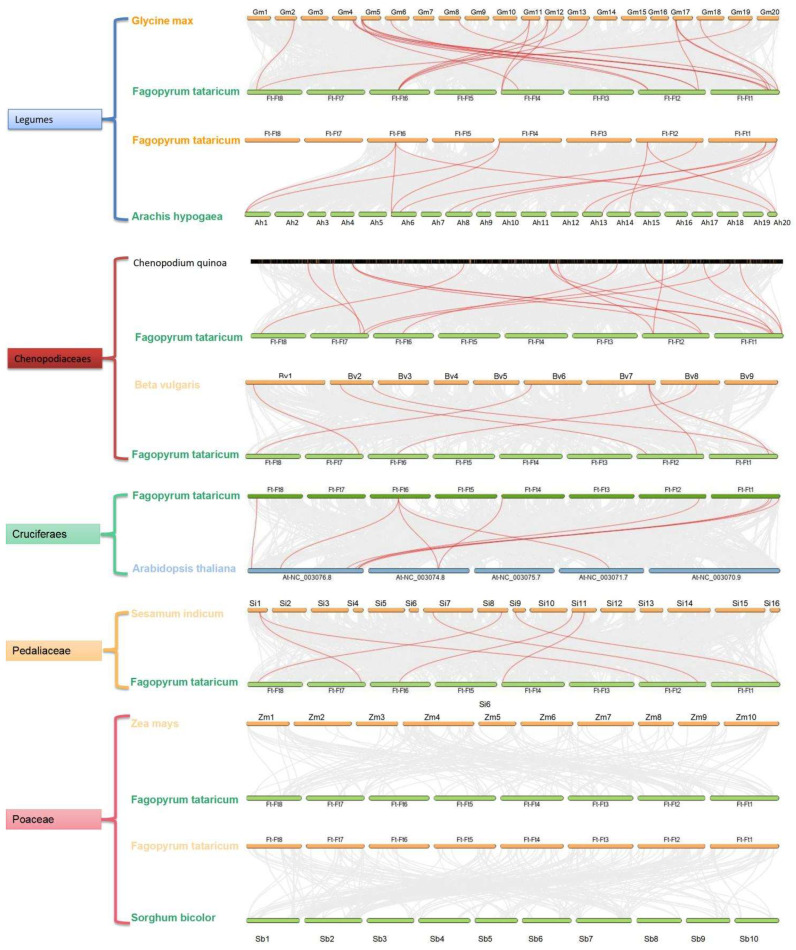
Synteny analyses of *HDACs* between *Fagopyrum tataricum* and other five representative plant species. Gray lines in the background indicate the collinear blocks within *Fagopyrum tataricum* and other plant genomes, while red lines highlight the syntenic *HDACs* gene pairs.

**Figure 5 ijms-23-07622-f005:**
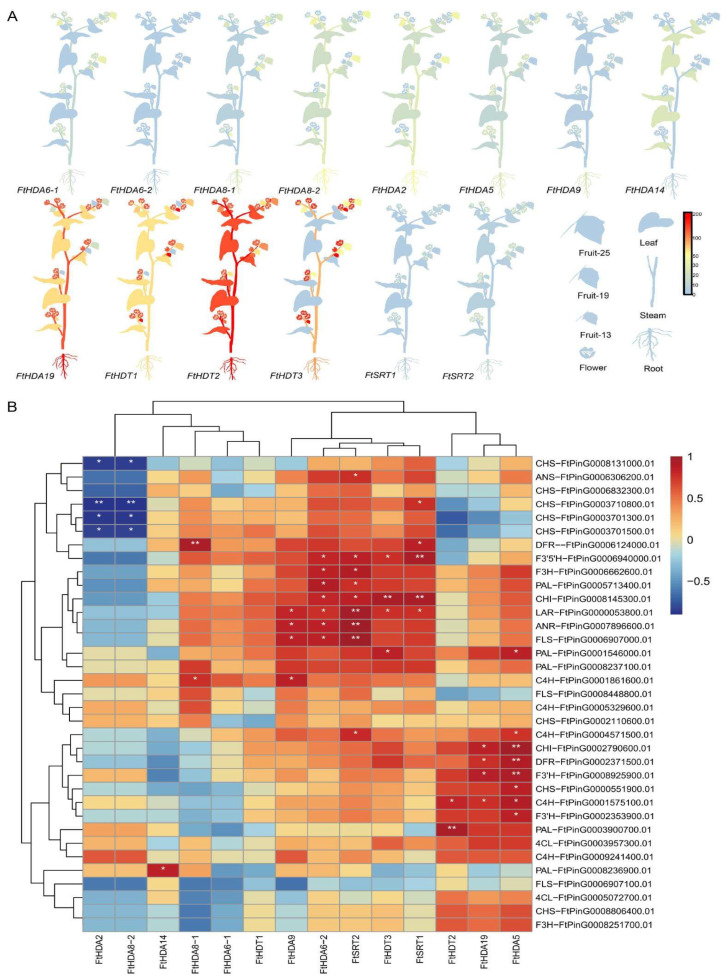
The tissue expression analysis of FtHDACs. (**A**) Expression patterns of *FtHDACs* genes in various Tartary buckwheat tissues. RNA-seq expression data corresponding to *FtHDACs* were retrieved from the Tartary Buckwheat Database (TBD) for further analysis. The RPKM (Reads Per Kilobase of exon model per Million mapped reads) values were transformed to log2(1 × 10^−6^). The expression in various Tartary buckwheat tissues is shown, including the root, stem, leaf, flower, and fruit_13, fruit_19, fruit_25. Blue to red, respectively, indicates a high to low expressional level. (**B**) The heatmap of correlation of *FtHDACs* and flavonoid related genes expression level is filtered. Red indicates the positive correlation, and blue indicates a negative correlation. Small white stars indicate significant associations (*: *p* ≤ 0.05, **: *p* ≤ 0.01).

**Figure 6 ijms-23-07622-f006:**
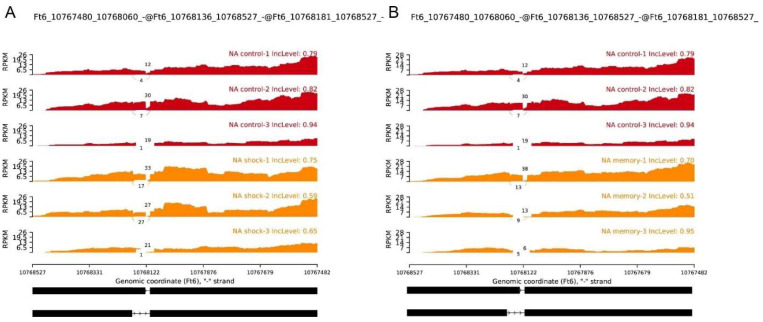
Alternative splicing of *FtHDA8-2* is associated with different low-temperature treatments. Sashimi plot indicating the average RNA-seq read density and splice junction counts for each genotype. (**A**) C group vs. S group. (**B**) C group vs. M group. C: living at room temperature always; S: not acclimated, directly exposed to 0 °C for 6 h; M: 4 °C for 6 h, then at room temperature for 18 h, repeated four times, and then placed at 0 °C for 6 h. C group (red) served as a negative control for S/M group (orange) enrichment.

**Figure 7 ijms-23-07622-f007:**
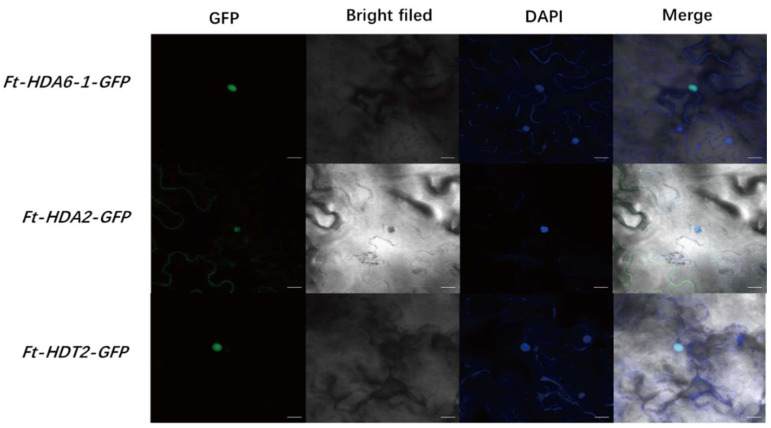
The subcellular localization of FtHDA6-1, FtHDA2 and FtHDT2. Bars = 20 μm.

**Figure 8 ijms-23-07622-f008:**
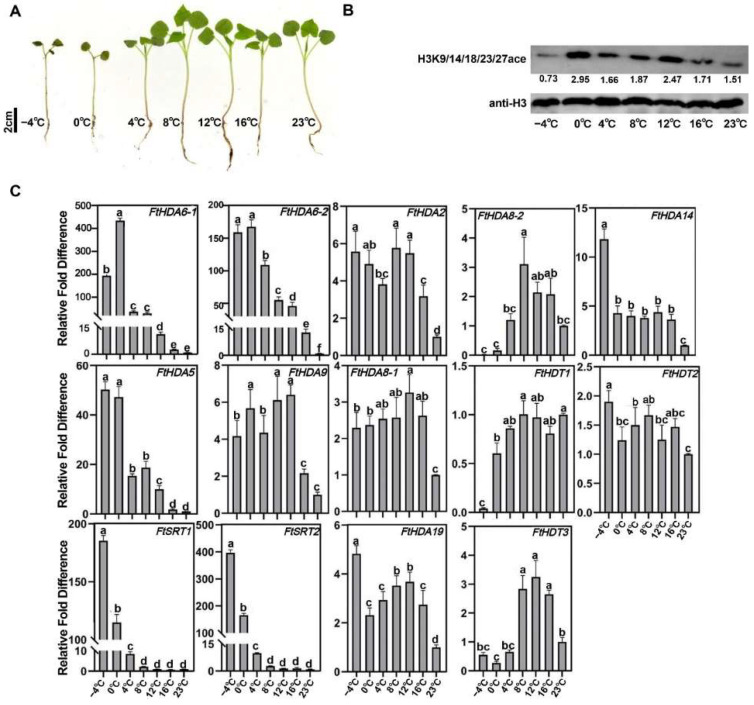
*FtHDACs* response to low temperature. (**A**) Phenotype of Dingku 1 displayed at different low temperatures. (**B**) The seedling of Dingku 1 treated in different low temperatures for 3 h then all recovered in 23 °C at 36 h. Western blot analysis of the change in the global histone 3 acetylation levels after low temperature treatments. Histone H3 was used as an equal loading control. All immunoblots were replicated three times for each sample from three independent experiments. (**C**) mRNA enrichment analysis of Dingku 1 in different cold treatments was performed. Data are presented as the means of three biological replicates (±SD). Different letters indicate significant differences (*p <* 0.05, Tukey’s test).

**Figure 9 ijms-23-07622-f009:**
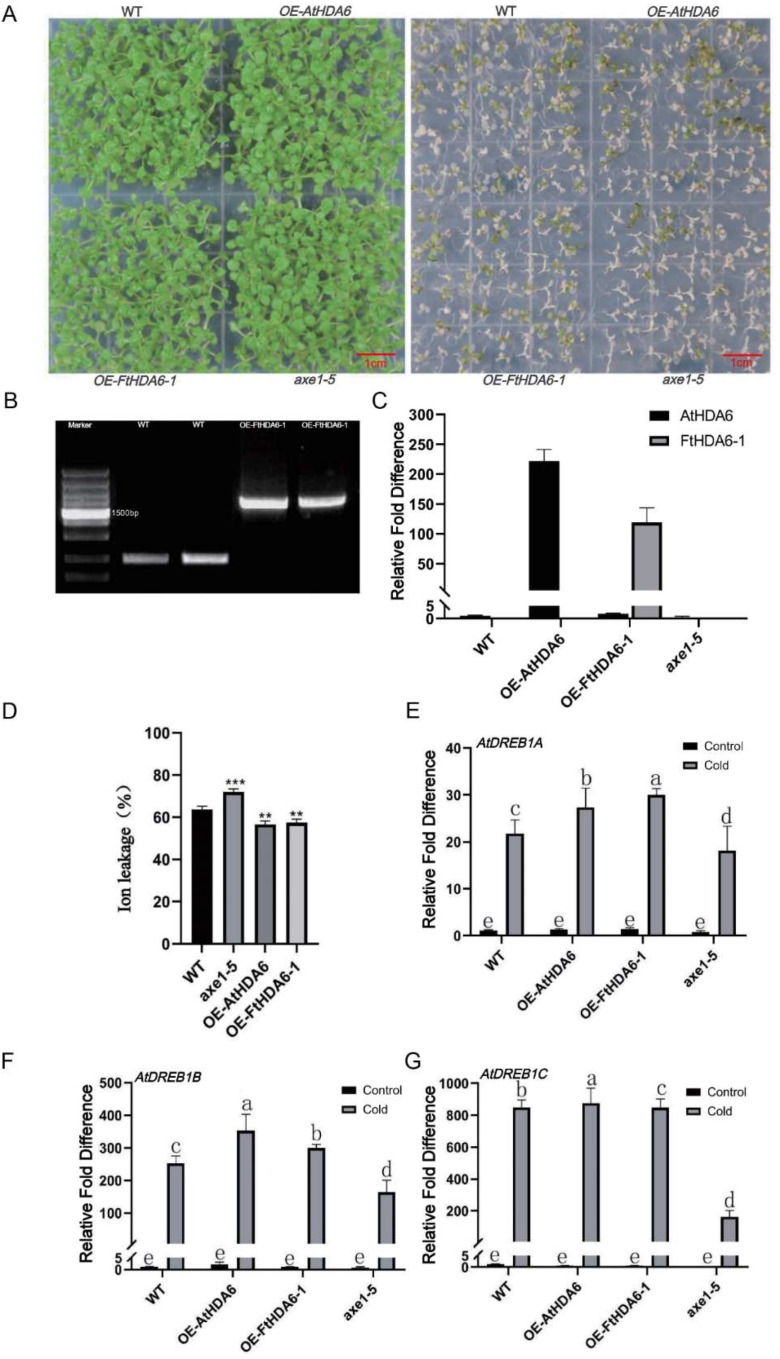
FtHDA6-1 positively regulates cold tolerance. (**A**,**D**) Phenotypes and ion leakage of seedling of WT, OE-*AtHDA6*, OE-*FtHDA6-1* and *axe1-5* under normal and cold stress conditions (−10 °C for 2  h, then recovery in 23 °C for 2 days). Data are presented as the means of three biological replicates (±SD). The asterisks indicate significant differences, one-factor ANOVA (*** p*  <  0.01, *** *p*  <  0.005). (**B**) Identification of transgenic *Arabidopsis thaliana*. (**C**) qRT-PCR identification the expression level of *HDA6* gene in WT, *axe1-5*, OE-*FtHDA6-1* and OE-*FtHDA6* genotypes. (**E**–**G**) Expression level of *CBFs* in different transgenic *Arabidopsis* lines under normal (the control) and cold stress conditions (0 °C for 3 days). Data are presented as the means of three biological replicates (±SD). Different letters indicate significant differences (*p <* 0.05, Tukey’s test).

**Figure 10 ijms-23-07622-f010:**
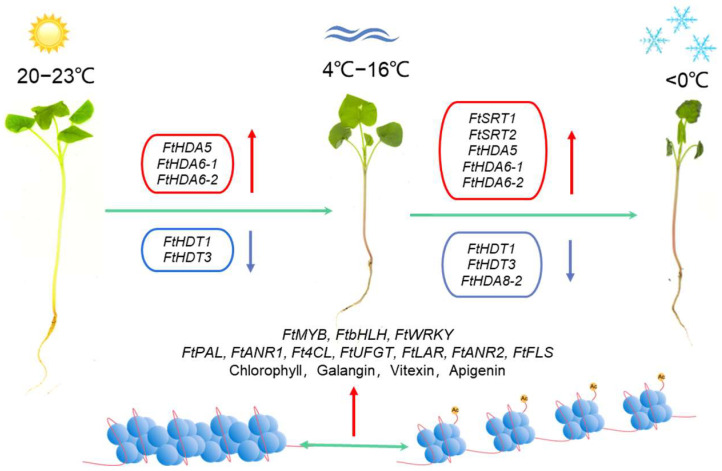
The proposed mechanism for the roles of *FtHDACs* in the regulation of low temperature and flavonoids biosynthesis in Tartary buckwheat. The level of histone acetylation affects chromatin conformation, and chromatin contraction and loosening regulate gene expression in response to low temperature and accumulation of flavonoids in Tartary buckwheat.

**Table 1 ijms-23-07622-t001:** Basic information of *HDACs* gene family in *Fagopyrum tataricum*.

Gene Name	Gene ID	CDS bp	Amino Acids aa	Molecular Weight	p*I*	Aliphatic Index	GRAVY
*FtHDA6-1*	FtPinG000668020	1428	475	53,654.75	5.39	70.78	−0.557
*FtHDA6-2*	FtPinG000582460	1458	485	54,108.42	5.74	68.72	−0.530
*FtHDA19*	FtPinG000705950	1509	502	56,189.23	5.08	73.51	−0.492
*FtHDA9*	FtPinG000324040	1293	430	49,188.5	5.06	79.56	−0.398
*FtHDA2*	FtPinG000223360	1092	363	39,980.06	8.44	98.87	−0.055
*FtHDA5*	FtPinG000450700	552	183	20,683.58	9.39	119.34	0.343
*FtHDA14*	FtPinG000269690	1299	432	46,834.56	6.10	91.23	0.035
*FtHDA8-1*	FtPinG000633580	849	282	30,216.81	5.34	83.01	−0.191
*FtHDA8-2*	FtPinG000633540	1149	382	41,256.46	5.28	88.85	−0.128
*FtSRT1*	FtPinG000031830	1431	476	52,748.52	9.35	91.09	−0.209
*FtSRT2*	FtPinG000180810	1398	465	51,761.15	8.50	91.96	−0.136
*FtHDT1*	FtPinG000141150	771	256	27,833.64	5.10	58.67	−0.972
*FtHDT2*	FtPinG000168350	645	214	23,549.38	4.11	66.07	−0.950
*FtHDT3*	FtPinG000710910	927	308	33,347.77	4.67	46.23	−1.202

## Data Availability

The transcriptome data have been deposited in National Center for Biotechnology Information’s Gene Expression Omnibus and are accessible through GSE138546.

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
