# Peer review of "Genome-Wide Analysis of the HDAC Gene Family and Its Functional Characterization at Low Temperatures in Tartary Buckwheat (Fagopyrum tataricum)"

_ijms, 2022, doi:10.3390/ijms23147622_

Round 1

Reviewer 1 Report

Hou et al. analyzed structures and functions of genes encoding HDAC from buckwheat. Although structural characterization of genes and their promoters were extensively performed, it was difficult to agree to author’s interpretation about the function of genes. In addition, there are two Figure 10, indicating requirement of the careful modification for the text.

Mainly, the first Figure 10 showed no significant phenotype relating cold stress tolerance in FtHDA6-1 and AtDHA6 transgenic lines. Moreover, expression levels of AtDREBs were not so different between WT and transgenic lines. However, authors said that “FtHDA6 improved freezing resistance, and induced high-profile expression of CBF genes”. Although integrated HAD genes were expressed highly in transgenic lines, no obvious phenotype was observed in my sense. Anyway, the line axe1-5 was used, there is no explanation about it.

The authors performed several lines of experiment; however, these were not integrated with each other to design experiments and make interpretations of the data. For instance, Figure 8 represented the expression profiles of genes under cold stress conditions, but alternative splice variants mentioned in Figure 6 were not focused on these experiments. Thus, interpretation of Figure 8 seems to be not appropriate. In addition, relationship between cold stress memory and gene expression was not addressed. In addition, Figure 7 showed the membrane localization of HDAs; however, there was no discussion about it to propose the functions of HDAs. The authors should re-consider about how your data must be interpreted with additional experiments.

Figure 3B is difficult to understand. Biosynthetic pathway of flavonoid with involvement of enzymes helps reader’s understanding.

Figure 2 showed the proposed cis-acting elements. Although many kinds of elements related to environmental stress response were found, there was no discussion about it with the possibility to participation of HDAs in cold stress and other stress response. Gene expression analyses are necessary.

The authors used “DREB” and “CBF” without any explanation. Are these the same? These abbreviations must be explained correctly.

The subtitle 2.8. in the Materials and Methods is not understandable. What is “and low”?

The words “2mm” in page 3 should be “2 mm “.

The sentence “The histones in the RPD3/HDA1 family consists of ----” in the Discussion is not understandable. Why do histone proteins can be consisted with zinc ions and amino acids?

There is no Data Availability information of the transcriptome data that must be deposited to the GenBank.

Reviewer 2 Report

In this work, Yukang Hou1, Qi Lu1, Jianxun Su, Xing Jin, Changfu Jia, Lizhe An, Yongke Tian, and Yuan Song, identified 14 HDACs genes, encoding a histone deacetylases, from Tartary buckwheat (Fagopyrum tataricum)  genome, in order to characterize their gene structure, motif composition, chromosome positions, their proteins and promoters, subcellular localization, gene expression pattern and alternative splicing events, as well as the roles of FtHDACs in the regulation of low temperature and flavonoids biosynthesis in Tartary buckwheat.

It should be noted that HDACs are involved in key plant process, including the biotic and abiotic stress responses in plants.

Initially, researchers searched for candidate genes that encode HDACs genes in the Tartary buckwheat genome, using pBLAST resource.

Further phylogenetic, feature domain analysis and so on in silico analysis showed that (i) FtHDAC family is divided into three subfamilies with a typical subfamily domain; (ii) all members contained one distinct motif for each subfamily. Further, the researchers conducted a search for cis-acting elements in FtHDAC gene promoters using the PLACE website. This in silico analysis revealed that the majority of the HcIAA promoters contained cis-acting elements involved in hormone response, stress response, functional control. Further, a comparative analysis of the relative expression of Fagopyrum tataricum FtHDAC and part of the flavonoid biosynthesis genes in roots, stems, leaves, flowers and three stages of fruit development was done using tartary buckwheat tissue transcriptome date. Additionally, alternative splicing of a number of transcripts during cold exposure of plants was evaluated using transcript data obtained earlier.

Several experiments were performed to establish of the effect of низких температур on the level of transcription of the some FtHDAC genes. The authors then attempted to determine subcellular location of the некоторых FtHDAC proteins (OE-FtHDA6-1, OE-FtHDA19, OE-FtHDA2, OE-FtSRT1 и OE-FtHDT2) and to evaluate how the expression OE-FtHDA6-1 gene in transgenic A.thaliana plants correlates with resistance of transgenic plants к низким температурам, за счет оценки уровня транскрипции генов транскрипционных факторов AtDREB1A, At-DREB1B и AtDREB1C у трансгенных линий растений, растений дикого типа и мутантной линии axe1-5 A.thaliana.

Major conclusions by the authors include (i) the 14 HDACs genes, encoding a histone deacetylases, were identified in Tartary buckwheat (Fagopyrum tataricum)  genome genome; (ii) the HDAC gene promoters contain several well-identified response elements including the hormone response, stress response, functional control; (iii) the some HDAC gene have preferential transcription patterns; (iv) FtHDACs showed expression in various tissues/organs in low-temperature environments; (v) the FtHDAC gene family might be involved in cold responses (FtHDA8-2 и FtHDT1 играют негативную регулирующую роль в Dingku 1 при низкотемпературном стрессе).

The experiments seem to be performed in sound techniques and presented in a proper manner.

The topic of this work is interesting; however, there are some comments to manuscript:

Major comments:

1.    The authors' main results are based on the analysis of transcriptomic data to assess the differential expression of FtHDACs and part of the flavonoid synthetic genes in different organs. It is necessary to indicate and briefly describe the basis of the method by which the differential expression of target genes was calculated. Additionally, in the "Materials and Methods" section, you must specify the data (Accession ID) on the placement of primary data in an open database - Tartary Buckwheat Database (TBD) (http://shu-juku.zuotukeji.net/). It is also necessary to indicate the source of the transcriptome and metabolome data of the three cold test groups that were used for the low-temperature resistance analysis of Dingku 1

2.    Regarding the comparative analysis of axe1-5 mutant plants, transgenic plants expressing the FtHDA6-1 or AtHDA6 gene. The manuscript states that “the FtHDA6-1-pCAMBIA1300-eGFP construct was inserted into Agrobacterium tumefaciens GV3101 cells that were used to transform transgenic Arabidopsis plants by flower dipping” (Materials and Methods section) and “To further investigate the function of FtHDAC, we constructed overexpression lines OE-AtHDA6 and OE-FtHDA6-1, and they have been identified. Subsequently, seedlings of four genotypes were frozen at -10°C for 2 hours and then restored at room temperature for 2 days" (Results section), however, there is no detailed description of how the transgenic plant lines were obtained and selected, nor is a brief characterization of axe1-5 mutant plants presented. Namely, there is no information on (i) how the target gene was cloned into the expression vector and what the fusion with the egfp gene (vector FtHDA6-1-pCAMBIA1300-eGFP) was used for; (ii) which selectable marker gene the vector contains; (iii) homozygous plant lines were obtained and their number; (iv) whether synthesis of the target protein has been proven in transgenic lines. These data are extremely important for selecting a line of transgenic plants and including them in subsequent experiments.

3.    When the authors describe the results of subcellular localization of the FtHDACs as well as the creation of transgenic plants expressing the FtHDA6-1, readers may find it interesting whether what criteria were used for the FtHDA6-1, FtHDA2, FtHDA19, FtHDT2 and FtSRT1 that were selected for further characterization.

4.    About the results regarding the subcellular localization of the FtHDA6-1, FtHDA2, FtHDA19, FtHDT2 and FtSRT1 hybrid genes fused with the reporter GFP gene. The authors indicate “To determining the location of FtHDACs functions, we constructed a vector expressing translational GFP-FtHDA6-1, GFP-FtHDA2, GFP-FtHDA19, GFP-FtHDT2 and GFP-FtSRT1 fusion proteins, which were infiltrated into tobacco leaves with Agrobacterim tumefaciens strain GV3101.  Through subcellular localization and RT-PCR analysis, the results showed all five genes were located in the nucleus. Additionally, except for FtSRT1, which only showed fluorescence signals in the nucleus, four other genes also had signals on the cell membrane.” However, the presented photographs (Fig. 7) do not allow us to judge so unequivocally that the hybrid genes were localized in the nucleus. First of all, there is no corresponding marker that would indicate a reliable localization of gene products in the cell nucleus (see the review by Alexander A. Tyurin et al.. Transient gene expression is an effective experimental tool for the research into the fine mechanisms of plant gene function: advantages, limitations, and solutions Plants 2020, 9, 1187; doi:10.3390/plants9091187). It should be explained how the results of RT-PCR analysis indicate that “all five genes were located in the nucleus”. Additionally, to assess subcellular localization, the authors used a very non-standard fusion, namely, obtaining constructs in which the reporter protein is located in the N-terminal region (construct names - GFP-FtHDA6-1, GFP-FtHDA2, GFP-FtHDA19, GFP-FtHDT2 and GFP-FtSRT1). As a rule, with extremely rare exceptions, the leader signals of subcellular localization are located in the N-terminal region, which, in the case of the authors, will not be available for the manifestation of its functional activity. Or are there data on the localization of such leader sequences in FtHDACs proteins in the C-terminal region?

5.    The results of the analysis of alternative splicing and transgenic plants are discussed very sparingly. It should be clarified what priority data were obtained in the course of the studies.

Minor comments:

1.    Check Subsection 2.2. “The secondary protein structure was performed by PRABI (http://www.prabi.fr/). An automated protein structure building was conducted by the Robetta (https://robetta.bakerlab.org/) program.” and Subsection 2.3. “The secondary protein structure was performed by PRABI (http://www.prabi.fr/). An automated protein structure building was conducted by the Robetta (https://robetta.bakerlab.org/) program.” Completely repeated.

2.    It should be clarified which method is used to predict the 3D structures that are presented in (Figure1B). Method of molecular homology or another? If the molecular homology method is used, it is necessary to indicate the ID of the proteins that were used to build the structures.

3.    There is no description of how the related expression level is calculated. The data in Fig. 8 and 10 are the values - ΔCT? Or 2- ΔΔCt? Please describe it in the "figure legends" or "Material and Methods".

4.      In the drawings Relative expression levels of FtHDACs genes, the same axes should be represented - Relative Fold Difference or Relative Fold Difference (2-ΔCt)

5.    Check the нумерацию figures in the manuscript: so, figure 10 presents twice and Figure 9 отсутствует (Section Results and Discussion)

Round 2

Reviewer 1 Report

The new version of the manuscript was well revised according to reviewer's comments. However, it is still problem that any visible phenotype and  induction of CBF gene expression was not observed in overexpressors. Although these results provide negative impression of the function of the HDA genes in cold stress response, the authors recognized it and tried to obtain non-visible phenotype. Thus, this manuscript is recognized as a report of the possible involvement of HDA in cold stress response, which must be evaluated further in near future.

Reviewer 2 Report

After reviewing the resubmitted manuscript and the authors' responses to my comments, the following should be noted: (i) the authors satisfactorily responded to all the questions and comments I have raised; (ii) the comments and questions posed by me has been clarified in manuscript.